# Differences between Survival Rates and Patterns of Failure of Patients with Lung Adenocarcinoma and Squamous Cell Carcinoma Who Received Single-Fraction Stereotactic Body Radiotherapy

**DOI:** 10.3390/cancers16040755

**Published:** 2024-02-12

**Authors:** Tyler V. Schrand, Austin J. Iovoli, Neil D. Almeida, Han Yu, Nadia Malik, Mark Farrugia, Anurag K. Singh

**Affiliations:** 1Department of Radiation Medicine, Roswell Park Comprehensive Cancer Center, Buffalo, NY 14263, USA; tschran@bgsu.edu (T.V.S.); austin.iovoli@roswellpark.org (A.J.I.); neil.almeida@roswellpark.org (N.D.A.); nadia.malik@roswellpark.org (N.M.); mark.farrugia@roswellpark.org (M.F.); 2Department of Chemistry, Bowling Green State University, Bowling Green, OH 43403, USA; 3Department of Biostatistics and Bioinformatics, Roswell Park Comprehensive Cancer Center, Buffalo, NY 14263, USA; han.yu@roswellpark.org

**Keywords:** SBRT, lung cancer, NSCLC, radiotherapy, radiation

## Abstract

**Simple Summary:**

Stereotactic body radiotherapy (SBRT) offers a highly conformal tumoricidal dose of radiation and is an effective treatment for early stage non-small cell lung cancer (NSCLC). Single-fraction SBRT (SF-SBRT) produces cost-effective and comparable outcomes to hypofractionated SBRT regimens. There is a limited understanding of patient outcomes after SF-SBRT according to the histology of their early stage NSCLC. We investigated the survival and patterns of failure of patients with adenocarcinoma (ADC) and squamous cell carcinoma (SCC). Out of the 292 eligible patients, 174 and 118 patients were diagnosed with ADC and SCC, respectively. Patients with ADC were found to be significantly more likely to experience a distant failure compared to patients with SCC. No significant differences were found in overall survival, disease-free survival, local failure, or nodal failure based on histology.

**Abstract:**

We investigated the survival and patterns of failure in adenocarcinoma (ADC) and squamous cell carcinoma (SCC) in early stage non-small cell lung cancer (NSCLC) treated with single-fraction stereotactic body radiation therapy (SF-SBRT) of 27–34 Gray. A single-institution retrospective review of patients with biopsy-proven early stage ADC or SCC undergoing definitive SF-SBRT between September 2008 and February 2023 was performed. The primary outcomes were overall survival (OS) and disease-free survival (DFS). The secondary outcomes included local failure (LF), nodal failure (NF), and distant failure (DF). Of 292 eligible patients 174 had adenocarcinoma and 118 had squamous cell carcinoma. There was no significant change in any outcome except distant failure. Patients with ADC were significantly more likely to experience distant failure than patients with SCC (*p* = 0.0081). In conclusion, while SF-SBRT produced similar LF, NF, DFS, and OS, the higher rate of distant failure in ADC patients suggests that ongoing trials of SBRT and systemic therapy combinations should report their outcomes by histology.

## 1. Introduction

Lung cancer is the leading cause of cancer-related mortality nationwide [1]. Approximately 230,000 individuals in the United States are diagnosed annually with lung cancer, resulting in nearly 135,000 deaths annually [2]. Lung cancer classifications are derived from the World Health Organization, and approximately 85% of all lung cancers are non-small cell lung cancer (NSCLCs). Adenocarcinoma histology accounts for approximately 50% of NSCLC diagnoses, followed by squamous cell carcinoma at 30% [1].

For medically operable patients with early stage lung cancer, surgical intervention is the preferred treatment modality. For patients who are medically inoperable or decline surgery, stereotactic body radiation therapy (SBRT) provides a highly conformal, tumoricidal radiation dose to thoracic tumors [3]. SBRT can be applied in one to five treatments, as well as hypofractionated regimens, for patients with early stage node-negative NSCLC who are not candidates for surgical intervention [4]. Prospective SBRT trials in medically inoperable and select operable patients with early stage NSCLC demonstrate a 3-year overall survival ranging from 43% to 95% with loco-regional control of up to 98% at 3 years [5].

The expediency of single-fraction SBRT was particularly highlighted during the coronavirus disease 2019 (COVID-19) outbreak, during which single-fraction SBRT was utilized to a higher degree to treat NSCLC patients [6,7,8]. The fewer treatment fractions correspond to an approximately 40% lower cost associated with single-fraction SBRT compared to the more traditional NSCLC fractionation of three fractions [6]. The adoption of single-fraction SBRT for NSCLC has still been relatively limited due to a multitude of factors, including a lack of understanding about its toxicity and efficacy, as well as a paucity of research pertaining to SBRT outcomes based upon the histological subtypes of NSCLC [9].

The potential impact of histological subtypes of NSCLC, either adenocarcinoma (ADC) or squamous cell carcinoma (SCC), on SBRT outcomes remains unclear. Kita et al. reported that the risk of LR after SBRT was higher for SCC than for ADC after applying a four-fraction regimen [10]. Additionally, Woody et al. found that SCC patients were significantly more likely to experience LR after employing five-fraction SBRT [11]. We aimed to investigate the utility of a single-fraction SBRT regimen for NSCLC in order to further our understanding of how to optimize SBRT treatments on the basis of a patient’s histology.

## 2. Materials and Methods

### 2.1. Study Population

This study was a review of a single institutional database of patients who received thoracic SBRT between September 2008 and February 2023. Eligibility was based on the following criteria: (1) biopsy-confirmed lung cancer; (2) early stage tumor (<IIB overall based on 8th TNM classification) [12]; (3) single-fraction SBRT; (4) definitive intent of treatment; (5) no previous radiation treatment of the lungs of any form; and (6) primary lung cancer. A total of 353 patients fit these criteria. Thirty-two patients were excluded due to a lack of biopsy, or a diagnosis of NSCLC not otherwise specified (i.e., poorly differentiated carcinoma). Twenty-nine patients were excluded due to a follow-up time of less than 6 months. In the 292 patients included in the review, 174 and 118 patients were diagnosed with ADC and SCC, respectively. Our patient selection process is modeled in Figure 1. Data were collected under the EDR 171710 protocol, which was approved by the institutional review board at Roswell Park Comprehensive Cancer Center. The Strengthening the Reporting of Observational Studies in Epidemiology (STROBE) reporting guideline was followed.

### 2.2. Clinical Evaluation and Follow-Up

In order to determine candidacy for thoracic SBRT, clinical evaluation and workup was utilized, following institutional guidelines and physician expertise. Per the ASTRO guidelines, patients who had declined surgical intervention or were deemed inoperable were considered for SBRT [13]. The thoracic surgeon had the authority to determine candidacy for surgical resection. A viable surgical candidate is a patient who is deemed medically fit to undergo a wedge resection or more-intensive surgery. Complex cases were discussed at a multidisciplinary case conference discussion. All patients were diagnosed with cN0 lung cancer, according to the Eighth Edition of the TNM Staging of Lung Cancer [14]. The presence of benign lymph nodes was determined by positron emission tomography with diagnostic computed tomography (PET/CT) imaging and/or endoscopic nodal sampling. We defined the central lung region as the region within 2 cm of the proximal bronchial tree [15]. For borderline cases, doses were deemed on an individual basis after a discussion of risks and benefits with the patient. After SBRT treatment, patients were evaluated in a follow-up three months after their radiation therapy was completed, with a diagnostic chest CT and by repeat chest imaging every 3–6 months within the first year. One year post treatment, the imaging schedule was changed to every 6 months. Image findings considered suspicious were further investigated with either a biopsy and/or PET/CT as deemed necessary.

### 2.3. Treatment Planning

Patients were treated with SF-SBRT as described by RTOG 1915 or RCPI-124407; a small number of patients were enrolled on both [16,17]. Patients underwent CT simulation in the supine position with arms above their head, using a thoracic Medical Intelligence BodyFIX^®^ immobilization system (Elekta, Stockholm, Sweden). Abdominal compression or respiratory gating was used to manage tumor motion, as described [17,18]. Non-coplanar three-dimensional conformal fields (3D-CRT) or volumetric modulated arc therapy (VMAT) were employed as dose delivery techniques. Heterogeneity corrections were used only for patients treated with intensity-modulated radiation therapy (IMRT). ROTG 0915 was utilized to provide normal tissue dose constraints [16]. Radiation treatment plans were generated and evaluated using Eclipse (Varian Medical Systems, Palo Alto, CA, USA). The majority of patients treated with SF-SBRT at Roswell Park CCI were treated with a dose of 27 or 30 Gy. The 27 Gy dose reflects heterogeneity corrections that were made following a new dose calculation algorithm implemented in 2017. Both doses deliver a similar biologically effective dose (BED), so they are felt to be equivalent [18].

### 2.4. Patient Data

A secure REDCap database was used to store pertinent clinicopathologic data extracted from the patients’ medical records [12,13]. Variables that were investigated included sex, race, performance status, medical history, smoking status, size, staging, tumor SUV, and operability status. The Karnofsky Performance Status (KPS) was used to quantify performance status [19].

Measured outcomes included overall survival (OS) and disease-free survival (DFS), which were coded as primary outcomes. Secondary outcomes included local failure (LF), nodal failure (NF), and distant failure (DF). Ipsilateral lung failures (IF) that did not meet the criteria of LF, NF, or DF were coded separately. Overall survival was defined as the time between SF-SBRT’s start date and either death or the last known follow-up. Disease-free survival was defined as the time between SF-SBRT’s start date and any tumor recurrence, last known follow-up, or death. Radiographic progression was determined at the discretion of the radiologist. In order to affirm tumor viability, either PET imagining with a similar uptake to the pretreatment staging PET, or a repeat biopsy that confirmed carcinoma could be used. Nodal failure was defined as tumor recurrence is any thoracic nodal station. Distant failure was defined as tumor recurrence either in the contralateral lung or extra-thoracically. Disease failures were evaluated in a multidisciplinary setting, and were based on radiographic findings. Biopsy results of metastatic sites were used if available. Toxicity data were not available.

### 2.5. Statistical Analysis

Estimates for OS and DFS were conducted using the Kaplan–Meier method, and the cumulative incidence of LF, NF, and DF was estimated using the Aalen–Johnson estimator with the patient death considered a competing risk. Statistical significance between ADC and SCC in each of the outcomes above was tested using the log-rank test. The Fine–Gray model was used for univariate analyses of the association between the histology and the outcomes. The Fine–Gray model was also used in a multivariate manner to control for possible confounding variables, which were found to be potentially associated with the histology (*p* < 0.1). The subdistributional hazard ratios, 95% Cls, and 3-year cumulative incidence rates were reported. Calculated *p*-values were two-sided with α = 0.05. Statistical analysis was conducted using R (version 4.2.0, R Project for Statistical Computing, Vienna, Austria) [20].

## 3. Results

### 3.1. Patient Population

Patient demographics, characteristics, and treatments are described in Table 1. All patients completed the planned SBRT treatment. The median follow-up time for all patients was 54 months (IQR 31–66). The median age at treatment was 76 years (IQR 69–81). The percentages of current and former smokers were higher in the SCC group (*p* = 0.0058). Additionally, the patients in the SCC group had a significantly higher pack year (*p* = 0.0057). There was no significant difference in tumor size between the two groups (*p* = 0.51), but the SCC group had a significantly higher SUV (*p* < 0.0001). Most patients were female (57%), but there was no significance between the two groups based on sex (*p* = 0.54). Overall, 84% of patients were deemed medically inoperable and 16% of patients were deemed medically operable but declined surgery. The first and second most common tumor locations were the right upper lobe and the left upper lobe, respectively, although there is no significant difference in location (*p* = 0.117). A dose of 2700 cGy was used to treat most patients (61.64%), which reflects the new dose algorithm implemented in 2017. Both the 2700 cGy and 3000 cGy doses deliver a similar biologically effective dose (BED). Almost half of the total patients were diagnosed with stage IA2 NSCLC (48.63%). In our cohort, 10 patients (~3.4%) were deemed to have central tumors within 2 cm of the proximal bronchial tree. Proximity to the chest wall was never factored into the decision to use SF-SBRT.

### 3.2. Outcomes

The treatment outcomes are described in Figure 2 and Figure 3 and Table 2. At the time of analysis, of the 292 total patients, 151 (52%) had died. Out of the 151 patients that died, 47 patients (31%) died due to the underlying pathology treated by SF-SBRT. The median OS was 68 months (IQR 39–112), and the median DFS was 48 months (IQR 29–91). The median OS (95% confidence interval) for ADC and SCC was 43.6 (35.1–54.7) months and 39.3 (32.1–48.2) months, respectively. For disease-free survival, the median values for ADC and SCC were 30.0 (26.3–45.7) and 33.8 (25.2–43.5), respectively. Figure 2A,B show the survival curves for overall survival and disease-free survival, respectively, with both charts separated by histology.

At the time of analysis, a total of 77 (26%) patients had experienced disease progression. The leading cause of disease progression was distant failure (DF), which was experienced by 46 (16%) patients. The second and third most common types of disease progression were lymph node metastasis (27 events, 9%) and local failure (24 events, 8%), respectively. Table 2 shows the 3-year survival rates and cumulative incidence rates of the measured outcomes, along with the univariate *p*-values. Recurrences were diagnosed via either biopsy, PET findings, CT findings, or at the discretion of the radiologist. Figure 3A–C display the cumulative incidence curves of local failure (LF), lymph node failure (NF), and distant failure (DF), respectively. Each chart separates the incidences by histology (ADC vs. SCC).

### 3.3. Analysis of Outcomes

Upon univariate analysis, there was found to be no significant difference between the two histologies in terms of OS, DFS, LF, or NF (*p* = 0.51, 0.81, 0.12, and 0.60, respectively). However, patients with ADC were found to be significantly more likely to experience a DF than patients with SCC (*p* = 0.0081) in the univariate analysis. Upon multivariate analysis, both histology and SUV were found to be significant predictors of a DF outcome (*p* = 0.0021 and 0.02, respectively). Smoking status and pack years were not found to be significant predictors of DF. The complete multivariate analysis of distant failure is tabulated in Table 3, with the subdistributional hazard ratio (SHR), 95% confidence intervals, and *p*-values listed.

## 4. Discussion

We found no significant differences in the 3-year outcomes (LF, NF, DFS, OS) of NSCLC patients treated with SF-SBRT, except for a significantly increased DF in ADC versus SCC (22 % vs. 8%, *p* = 0.0081). In a multivariate analysis, histology and SUV were found to be significant predictors of distant failure, with hazard ratios of 0.26 and 1.08, respectively.

This absence of the impact of histology on LF with SF-SBRT contrasts with prior studies that have shown that LF was significantly higher in SCC versus ADC treated with five-fraction SBRT. Woody et al. reported that the 3-year cumulative incidence of local failure for SCC was significantly higher, 18.9% versus 8.7% (*p* = 0.008), than for ADC following 50 Gray (Gy) SBRT in five fractions [11]. Similarly, Kita et al. found that SCC histology had a hazard ratio of 2.41 (*p* = 0.012) for local failure [10]. Kita et al. also reported a multivariate subgroup analysis based on tumor sizes ≤2.5 cm and >2.5 cm; the authors reported that in the >2.5 cm group, SCC was an independent factor for local failure (HR 2.61, *p* = 0.036) [10]. Our analysis demonstrated a median SCC tumor size of 1.9 cm with no significant change in local failure with increasing size, which is concordant with the findings of Kita et al. [10]. In addition, Baine et al. also found SCC to be a risk factor for local failure after SBRT for early stage NSCLC [20].

A number of prior studies have demonstrated that SCC is more radiosensitive than other histologies [21,22,23,24]. Our study suggests relatively a similar radiosensitivity between SCC and ADC in the lung, reflected in the similar local failure rates in our patient population. More importantly, our results suggest that a therapeutic strategy for distal control is more important for cancer control in the ADC group than in the SCC group.

Therapeutic strategies designed to control distant metastases post SBRT, outlined by the ESTRO/ACORP guidelines for the treatment of NSCLC with SBRT, do not take the neoplastic histology into account [25]. In order to control the higher rates of DF in ADC that our study identified; we postulate that future studies will investigate the role of systemic therapeutics alongside SF-SBRT. Potential therapy regimens include utilizing SBRT with neoadjuvant immunotherapy [26,27,28]. Immunotherapy and SBRT have been shown to exhibit a synergistic affect when used together in the treatment of NSCLC [29,30]. Additionally, the NEOSTAR clinical trial found that monoclonal antibodies, when paired with chemotherapy, induce greater pathologic response rates when compared with chemotherapy alone in the treatment of NSCLC [31]. Future clinical trials could aim to evaluate the impact of such systemic regimens with SF-SBRT.

The radiobiological mechanisms contributing to differential distal failure in ADC due to SF-SBRT is unclear. Park et al. stated that single-fraction doses of higher than 10 Gy cause severe vascular damage to the tumor, which makes the tumor environment hypoxic, acidic, and deprived of nutrients when compared to a traditionally fractionated treatment regimen [32,33]. Future studies elucidating the differences in tumor biology based on histology could lead us to better understand these patient outcomes.

Data from multiple trials and analyses show no difference in NSCLC outcomes in terms of SF- versus fractionated-SBRT [9,16,29,34]. In addition, based on 2009 Medicare reimbursements, SF-SBRT for NSCLC was approximately 40% less expensive compared to three-fraction SBRT [6]. If confirmed in other analyses, this excellent local control of SCC by SF-SBRT, which is not maintained with multi-fraction regimens, may be added to the reduced cost and increased patient convenience as a reason to choose single-fraction SBRT. There are also data suggesting that SF-SBRT may have different patterns of failure than multi-fraction regimens [16,17]. Siva et al. reported that single-fraction SABR patients with primary renal cell carcinoma appear to be less likely to progress distantly or die of cancer compared to multi-fraction SABR [29].

Our study is limited by its retrospective nature and as it is a single-institution study. In addition, while this study incorporates NSCLC histology, it does not consider the pathological subtypes of adenocarcinoma. One subtype, lepidic pattern adenocarcinoma, has been noted to be indolent and less likely to metastasize when compared with other subtypes [35,36]. Furthermore, toxicity was not contemporaneously captured in our database and, therefore, is not quantitated. However, qualitatively, the toxicity of single-fraction SBRT (including in the chest wall) is quite low in our experience. This is consistent with prospective data [9,37].

## 5. Conclusions

In conclusion, the LF, NF, DFS, and OS of early stage NSCLC patients treated with SF-SBRT were examined, comparing between ADC and SCC. There was no significant change in local failure, but patients with ADC were significantly more likely to experience distant failure than patients with SCC. This study adds to the literature that examines treatment approaches based on the histological type of early stage NSCLC.

## Figures and Tables

**Figure 1 cancers-16-00755-f001:**
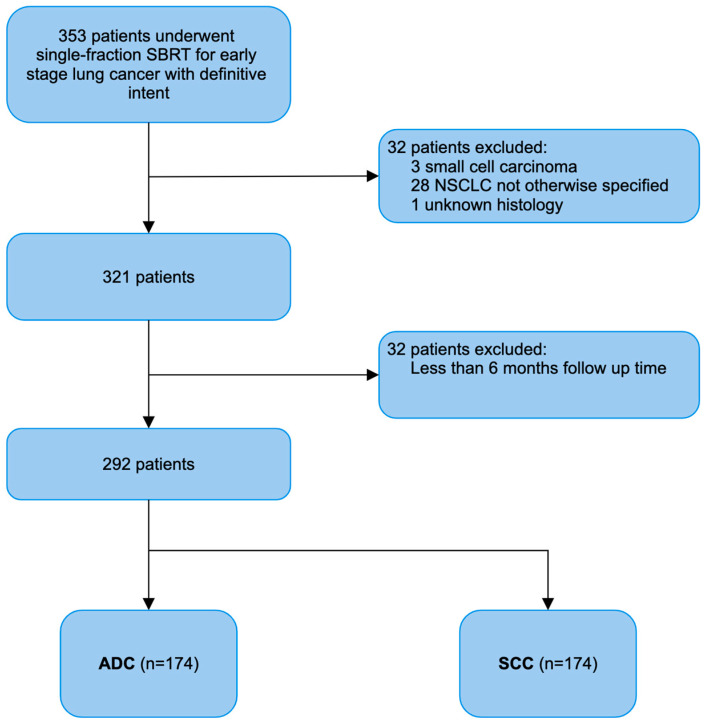
Patient selection flow chart.

**Figure 2 cancers-16-00755-f002:**
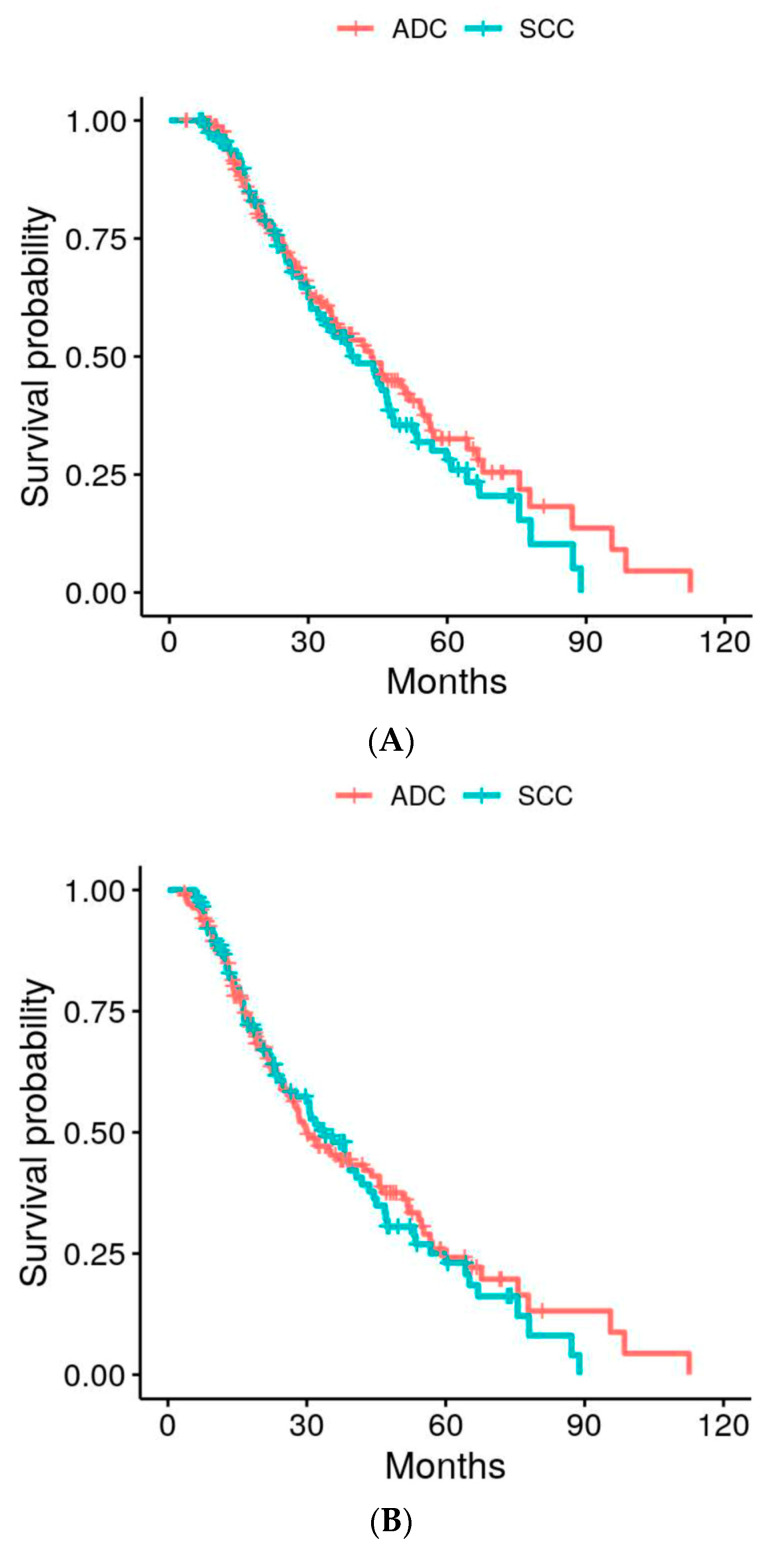
(**A**) Comparison of overall survival (OS) between patients with adenocarcinoma (ADC) and squamous cell carcinoma (SCC) *p* = 0.5168. (**B**) Comparison of disease-free survival (DFS) between patients with adenocarcinoma (ADC) and squamous cell carcinoma (SCC) *p* = 0.8055.

**Figure 3 cancers-16-00755-f003:**
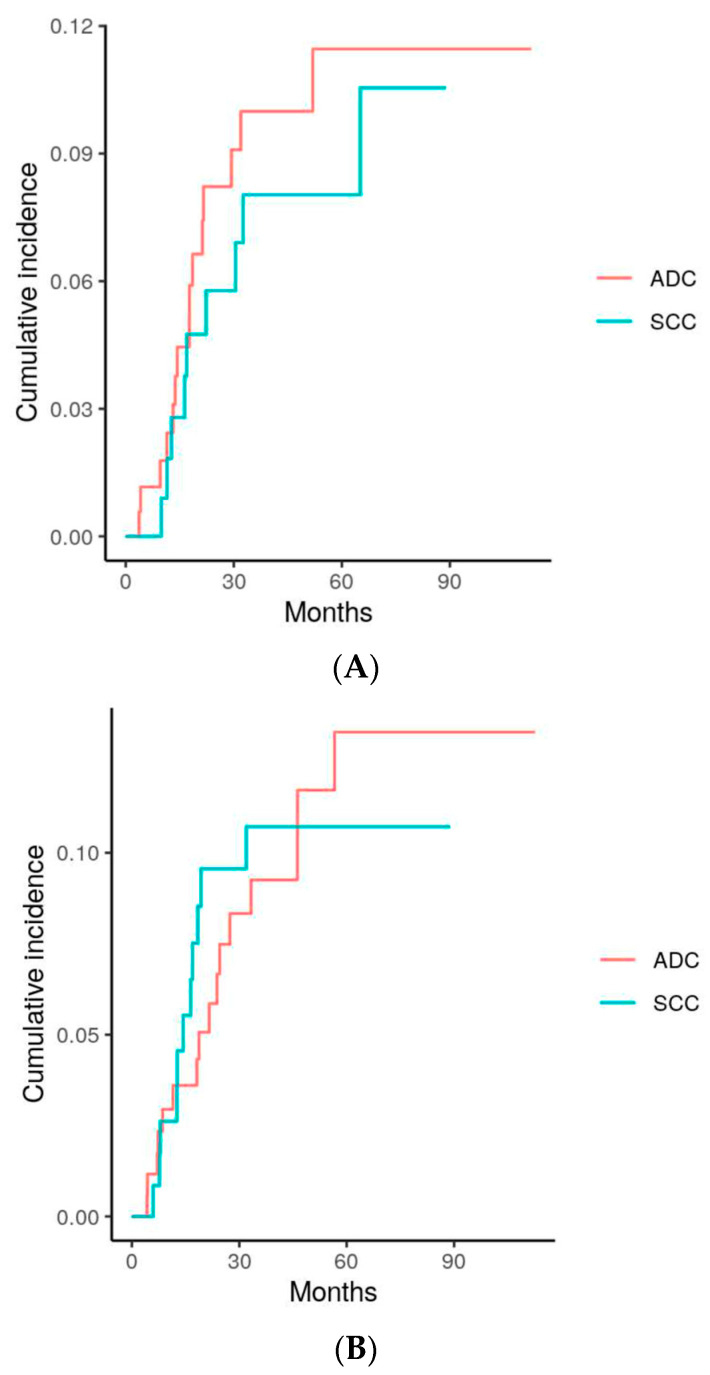
(**A**) Comparison of local failure (LF) between patients with adenocarcinoma (ADC) and squamous cell carcinoma (SCC) *p* = 0.6. (**B**) Comparison of lymph node failure (NF) between patients with adenocarcinoma (ADC) and squamous cell carcinoma (SCC) *p* = 0.95. (**C**) Comparison of distant failure (DF) between patients with adenocarcinoma (ADC) and squamous cell carcinoma (SCC) *p* = 0.0081.

**Table 1 cancers-16-00755-t001:** Patient demographics.

Characteristic	All (n = 292)	ADC (n = 174)	SCC (n = 118)	*p* Value
Sex				0.54
Male	125 (42.81)	66 (37.93)	59 (50.00)	
Female	167 (57.19)	108 (62.07)	59 (50.00)	
Median age, y	76.1 (69.4–81.1)	76 (67.9–80.6)	76.3 (71.6–80.5)	0.51
Race				0.313
White	265 (90.75)	156 (89.66)	109 (92.37)	
Black	14 (4.79)	11 (6.32)	3 (2.54)	
Other/unknown	13 4.45)	7 (4.02)	6 (5.08)	
KPS				0.9
≥80	183 (62.67)	110 (63.22)	73 (61.86)	
<80	109 (37.33)	64 (36.78)	45 (38.14)	
Smoking status				0.0058
Current	74 (25.34)	40 (22.99)	34 (28.81)	
Former	197 (67.47)	115 (66.09)	82 (69.49)	
Never	21 (7.19)	19 (10.92)	2 (1.69)	
Median pack years	50 (30–64.5)	44 (28–60)	50 (40–70)	0.0057
EBUS performed				0.414
Yes	216 (73.97)	49 (28.16)	91 (77.12)	
No	76 (26.03)	125 (71.84)	27 (22.88)	
Tumor site				0.117
Left upper lobe	79 (27.05)	55 (31.61)	24 (20.34)	
Left lower lobe	47 (16.10)	26 (14.94)	21 (17.80)	
Right upper lobe	91 (31.16)	49 (28.16)	42 (35.59)	
Right middle lobe	14 (4.79)	6 (3.45)	8 (6.78)	
Right lower lobe	60 (20.55)	38 (21.84)	22 (18.64)	
Other	1 (0.34)	0 (0.00)	1 (0.85)	
Median tumor size, cm	1.8 (1.3–2.5)	1.8 (1.3–2.5)	1.9 (1.3–2.45)	0.51
Median SUV value	5.4 (3.1–8.8)	4.2 (2.7–7.2)	7 (4.3–10.3)	<0.0001
SBRT dose				0.11
2700 cGy	180 (61.64)	114 (65.52)	66 (55.93)	
3000 cGy	109 (37.33)	58 (33.33)	51 (43.22)	
3400 cGy	3 (1.03)	2 (1.15)	1 (0.85)	
Overall stage				0.87
IA1	29 (9.93)	17 (9.77)	12 (10.17)	
IA2	142 (48.63)	86 (49.43)	56 (47.46)	
IA3	73 (25.00)	45 (25.86)	28 (23.73)	
IB	27 (9.25)	12 (6.90)	15 (12.71)	
IIA	11 (3.77)	7 (4.02)	5 (4.24)	

**Table 2 cancers-16-00755-t002:** 3-year survival rates and cumulative incidence rates for different outcomes.

Variable	All	ADC	SCC	*p*-Value
OS	0.556 (0.493, 0.626)	0.567 (0.486, 0.663)	0.541 (0.449, 0.653)	0.5168
DFS	0.464 (0.403, 0.534)	0.453 (0.375, 0.547)	0.48 (0.389, 0.592)	0.8055
LF	0.092 (0.055, 0.129)	0.1 (0.049, 0.15)	0.08 (0.026, 0.134)	0.6
NF	0.098 (0.06, 0.136)	0.093 (0.044, 0.142)	0.107 (0.047, 0.168)	0.95
DF	0.157 (0.111, 0.285)	0.216 (0.147, 0.285)	0.077 (0.025, 0.128)	0.0081

**Table 3 cancers-16-00755-t003:** SHR and multivariate analysis of distant failure.

Variable	SHR (95% CI)	*p*-Value
Histology (SCC vs. ADC)	0.26 (0.11,0.61)	0.0021
SUV	1.08 (1.01, 1.16)	0.02
Never Smoker (vs. Current Smoker)	1.66 (0.24, 11.46)	0.61
Former Smoker (vs. Current Smoker)	0.58 (0.28, 1.23)	0.16
Pack Years	0.96 (0.47, 1.97)	0.91

## Data Availability

The data presented in this study are available on request from the corresponding author. The data are not publicly available due to the private nature of the data.

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
