# Peer review of "Differences between Survival Rates and Patterns of Failure of Patients with Lung Adenocarcinoma and Squamous Cell Carcinoma Who Received Single-Fraction Stereotactic Body Radiotherapy"

_cancers, 2024, doi:10.3390/cancers16040755_

Round 1

Reviewer 1 Report

Comments and Suggestions for Authors

This article reports a monocentric serie of patients treated for early stage NSCLC with a single dose of sterotaxic radiotherapy. The aim was to compare the evolution of non-squamous and squamous cells carcinomas.

This is a retrospective study, but the patients included are homogeneous, with fairly identical radiotherapy doses, well-described technics, and a fairly homogeneous extension work-up, including  mediastinal exploration by EBUS.

The article is clear and well-written, with an interesting discussion.

There are some unanswered issues and necessary clarifications

1) We understand from the text that single-dose radiotherapy was systematic during the COVID period. But before and after, what were the criterias for single-dose versus 3 to 5 sessions?

2) some authors have described greater toxicities for tumors located centrally near the hilum and for those located very peripherally. it's likely that very central tumors were not included because a diagnosis was required, but what about very peripheral tumors close to the wall?

3) no mention is made of toxicity. this is not the subject of the article, but was it looked at?

4) On page 6, the OS is given at 68 months, which visually does not seem to correspond to figure 2. The same applies to the DFS.

5) There's an error in the name of the second curve: "A" instead of "B".

6) reference 18 is a bit curious, as it's an appendix to an article.

this article is interesting and deserves publication after a few corrections.

Author Response

  • We understand from the text that single-dose radiotherapy was systematic during the COVID period. But before and after, what were the criteria for single dose versus 3 to 5 sessions?
    • Thank you for the query. We deemed that the central lung region is a region defined as 2 cm within the proximal bronchial tree and has been proposed as the “No-Fly Zone.”  These patients predominantly received 5 fraction SBRT regimens at our institution. For borderline cases, doses were deemed on an individual basis after a discussion of risks and benefits with the patient. In our cohort, 10 patients (~3.4%) were deemed to have central tumors, with 60% of these central tumors receiving SF-SBRT during or after the COVID-pandemic. We have updated the methods & results section to reflect this.
  • Some authors have described greater toxicities for tumors located centrally near the hilum and for those located very peripherally. It's likely that very central tumors were not included because a diagnosis was required, but what about very peripheral tumors close to the wall?
    • Please see discussion above for #1. We have never considered peripheral tumors close to the chest wall to be contraindication for single fraction radiation. Qualitatively our institution has had quite low toxicity with single fraction SBRT. This is quantitatively supported by the prospective literature as reviewed by Bartl et al Cancers 2022.
  • No mention is made of toxicity. This is not the subject of the article, but was it looked at?
    • Qualitatively the toxicity of single fraction SBRT is quite low in our experience. This is consistent with prospective data (Bartl et al 2022, Sogono et al 2021). However, toxicity was not contemporaneously captured in our database and therefore is not quantitated. We have added this description of toxicity to our manuscript and updated the references accordingly.
  • On page 6, the OS is given at 68 months, which visually does not seem to correspond to figure 2. The same applies to the DFS.
    • The error was corrected. We appreciate the insightful comment.
  • There's an error in the name of the second curve: "A" instead of "B".
    • We have addressed this, thank you.
  • Reference 18 is a bit curious, as it's an appendix to an article.
    • The reference was changed to a more suitable reference, thank you.

Reviewer 2 Report

Comments and Suggestions for Authors

Stereotactic radiotherapy (SBRT) is a unique method of treating cancer and pathological lesions, which involves administering one or several (usually 1-5 fractions) large doses of radiation to the tumor area with a minimum volume of surrounding healthy tissues. Stereotactic radiosurgery is used as an element of radical, as well as palliative and analgesic treatment. In many cases, it is a reasonable alternative to the higher-risk classic surgical treatment. Treatment of non-small cell lung cancer (NSCLC) includes surgery, radiotherapy, chemotherapy, molecularly targeted drugs, immunotherapy and combination methods. The choice of the optimal treatment method for a given cancer should take into account all clinical and pathological features as well as the results of imaging, genetic and laboratory tests, as well as the patient's general condition and current symptoms of the disease. Single-fraction SBRT (SF-SBRT) provides cost-effective and comparable results to hypofractionated SBRT regimens.

The authors of this article examined the survival and failure patterns of early-stage adenocarcinoma (ADC) and squamous cell carcinoma (SCC) of non-small cell lung cancer treated with single-fraction stereotactic body radiotherapy (SF-SBRT) to 27-34 Gray. It was only found that patients with ADC were significantly more likely to suffer long-term failure than patients with SCC (p = 0.0081). It was confirmed that the treatment method should be based on the histopathological result of NSCLC.

The article supplements knowledge about the use of single-fraction SBRT, although there is no information about the toxicity of the treatment method. Please complete this data.

Author Response

Reviewer 2:

Stereotactic radiotherapy (SBRT) is a unique method of treating cancer and pathological lesions, which involves administering one or several (usually 1-5 fractions) large doses of radiation to the tumor area with a minimum volume of surrounding healthy tissues. Stereotactic radiosurgery is used as an element of radical, as well as palliative and analgesic treatment. In many cases, it is a reasonable alternative to the higher-risk classic surgical treatment. Treatment of non-small cell lung cancer (NSCLC) includes surgery, radiotherapy, chemotherapy, molecularly targeted drugs, immunotherapy and combination methods. The choice of the optimal treatment method for a given cancer should take into account all clinical and pathological features as well as the results of imaging, genetic and laboratory tests, as well as the patient's general condition and current symptoms of the disease. Single-fraction SBRT (SF-SBRT) provides cost-effective and comparable results to hypofractionated SBRT regimens.

The authors of this article examined the survival and failure patterns of early-stage adenocarcinoma (ADC) and squamous cell carcinoma (SCC) of non-small cell lung cancer treated with single-fraction stereotactic body radiotherapy (SF-SBRT) to 27-34 Gray. It was only found that patients with ADC were significantly more likely to suffer long-term failure than patients with SCC (p = 0.0081). It was confirmed that the treatment method should be based on the histopathological result of NSCLC.

The article supplements knowledge about the use of single-fraction SBRT, although there is no information about the toxicity of the treatment method. Please complete this data.

  1. We have updated the manuscript to reflect information about the toxicity. Qualitatively the toxicity of single fraction SBRT is quite low in our experience. This is consistent with prospective data (Bartl et al 2022, Sogono et al 2020). However, toxicity was not contemporaneously captured in our database and therefore is not quantitated. We have added this description of toxicity to our manuscript and updated the references accordingly.

Reviewer 3 Report

Comments and Suggestions for Authors

Thank you very much for this very interisting manucsript.

Just one comment: line 163 “The first and second most common tumor locations were the right upper lobe left upper lobe, respectively, although there is no significant difference in location (p = 0.117).” I think there is an “and” missing?

Author Response

Thank you very much for this very interesting manuscript.

Just one comment: line 163 “The first and second most common tumor locations were the right upper lobe left upper lobe, respectively, although there is no significant difference in location (p = 0.117).” I think there is an “and” missing?

  1. This has been appropriately fixed in the manuscript, thank you.

Reviewer 4 Report

Comments and Suggestions for Authors

The study design and analysis results are very appropriate and well presented. Given the large population for retrospective analysis, it has reliable statistical significance for evidence-based medicine.

Author Response

The study design and analysis results are very appropriate and well presented. Given the large population for retrospective analysis, it has reliable statistical significance for evidence-based medicine.

  1. Thank you for your kind commentary. We are thankful for the opportunity to submit our findings.